# Physiological Parameters Monitored on Bottlenose Dolphin Neonates (*Tursiops truncatus*, Montagu 1821) over the First 30 Days of Life

**DOI:** 10.3390/ani11041066

**Published:** 2021-04-09

**Authors:** Barbara Biancani, Guillermo J. Sánchez-Contreras, Stefano Furlati, Francesco Benaglia, Carmen M. Arija, Claudia Gili

**Affiliations:** 1Costa Edutainment S.p.A., Via Ascoli Piceno 6, 47838 Riccione, Italy; frullo72@gmail.com; 2Mediterraneo Marine Park, Coast Rd, NXR9038 Bahar ic-Caghaq, Malta; francesco.benaglia@anicura.de; 3AniCura Kleintierspezialisten Augsburg, Max-Josef-Metzger-Straße 9, 86157 Augsburg, Germany; 4Sea Wolves, Gorrión 64, 28019 Madrid, Spain; arija@seawolves.es; 5Stazione Zoologica Anton Dohrn, 80122 Naples, Italy

**Keywords:** bottlenose dolphin, *Tursiops truncatus*, parturition, neonate, observations, apnea, nursing activity, body weight, blood parameters

## Abstract

**Simple Summary:**

Bottlenose dolphins have been bred in controlled environments for more than 60 years. Health issues in newborn dolphins are difficult to diagnose; therefore, early detection is essential to allow prompt intervention. This study presents the retrospective results of observations and veterinary examination of 13 dolphin neonates born in three different European facilities from 2010 to 2018. Valuable information regarding partum observations and neonatal apnea, nursing activity, morphometric measurements, and blood parameters were collected in the first 30 days of life. These data are reported to highlight their relevance in evaluating bottlenose dolphin calves’ survivability and development. Considering the paucity of literature available on dolphin neonates, this long term study provides a large set of clinical parameters and a summary of the relevant conditions that could induce medical intervention. All the reported information can be useful for the colleagues working in the marine mammal veterinary field.

**Abstract:**

Bottlenose dolphins (*Tursiops truncatus*) have been bred under human care for more than 60 years. Calves up to 30 days of life have presented the highest mortality rate, although comparable data for free-ranging neonates within this age group is not available. Husbandry measures to increase survivability have therefore been constantly improved. This work shows the results of a structured veterinary program that established the procedures to collect relevant physiological parameters on 13 calves during their first 30 days of life. Standardized observation protocols facilitated statistical analysis of the respiratory pattern, nursing, morphometric measurements and bloodwork. These allow early detection of health issues. Healthy neonates had longer apnea duration, despite the shape and size of the facility. The nursing pattern showed that successful calves started nursing 3 to 18 h postpartum. Although with different individual patterns, a steady increase in suckling time occurs during the first 24 h of life. The healthy neonates grew 0.428 ± 0.102 kg per day and the complete blood count profile, glucose, iron, blood urea nitrogen (BUN), total protein, Alanine aminotransferase (ALT), Aspartate aminotransferase (AST), Gamma-glutamyl transferase (GGT), creatinine and electrolytes values considered normal for healthy calves are provided. Furthermore, cholesterol, triglycerides, α-amylase, lipase, magnesium and cortisol are reported for the first time for such young calves. A list of indications for prompt intervention is included.

## 1. Introduction

Medical intervention on mammal calves represents a regular veterinary procedure. It is carried out for domestic animals, although rarely indicated in zoo species for which direct contact with keepers is generally avoided [1]. The handling of live cetaceans’ calves was not considered an option up to 20 years ago, due to the lack of knowledge on manual safe restraint and stress avoidance [2] as well as technical limitations. These constraints were overcome with the introduction of lifting platforms that in the last 10–15 years have drastically improved the husbandry of bottlenose dolphin calves (*Tursiops truncatus*) under human care. Proactive physical handling has in fact facilitated the medical approach and animals’ population management [3]. 

Neonatal mortality of bottlenose dolphins has been documented both in the wild and in captivity over the last four decades [3,4,5,6,7,8,9,10,11,12]. The highest mortality rate was observed during the first month of age compared to any other age category in captive settings. Some of the causes for early deaths among dolphins of this age range remain unknown, although trauma, infection and “failure to thrive” (as extensively described in the literature) are considered to be among the primary explanations for dolphin mortality within the first 30 days of life [3,4,5,6,7,8,9,10,11,12,13]. Infection and failure to thrive are most probably related to the lack of immunity competence in the calf. In dolphins, due to the presence of diffuse epitheliochorial placentation, the maternal immunity is passed on to the calf through the colostrum [14]. For this reason, it is of the utmost importance that the calf receives the colostrum within the first 24–48 h of life [15].

The 2013 Report on *T. truncatus* of the European Association of Zoo and Aquaria Ex situ Program (EAZA EEP) [11] and the more recent long-term EEP management plan for the species [12], showed how the neonatal mortality of dolphins under human care in Europe has steadily decreased decade by decade in the last 40 years, from 77%, reaching a historical minimum of 33% (26/80) for the period 2011–2015. Jakkola et al. in 2019 [16] show that in United States, dolphin survival rates have increased significantly. Median life expectancy has more than tripled over the last few decades. Life expectancy is at least as high as in the wild populations. Nevertheless, the comparison with the survivability of young calves in the wild is not available, as neonatal mortality events are usually unreported or remain unnoticed [8,17,18], rendering the levels difficult to determine with certainty [16,19,20]. 

Three European zoological parks with different dolphin handling experience are considered here: Acquario di Genova (ADG), Mediterraneo Marine Park (MMP) and Oltremare (OLT). ADG opened in 1992, has housed dolphins since 1993 and had previous successful experience with dolphin calves in 1993 and in 1996, when “hands off” was the standard procedure and calves were not captured for medical check-ups. MMP opened to the public in 1997, and had dolphin births for the first time in 2010. OLT opened in 2004, and had several calves born by 2008, which all died within the first nine days of life due to bacterial infections. Considering the history of these facilities and the lack of handling experience, and highlighting that neonatal mortalities occur during the first month of life, prompt intervention after birth was considered essential [3,5,9,10,13,16,21]. In 2010 a combined protocol was created by veterinarians and biologists to schedule all the procedures and observations to occur prior to, during and after birth, in order to intervene with the calf in due time. It also prescribed physical handling and examination, blood and biometric data collection, prophylactic administration of long lasting antibiotic treatment within the second day of life. 

The aim of this paper is to present a summary of the results of the structured veterinary program that started in 2010. This program allowed the collection of intrapartum and postpartum data from 13 dolphin neonates during their first 30 days of life in order to foster calves’ survivability. Objective parameters, such as a timeline for specific events during the intrapartum phase, neonatal apnea duration, nursing frequency, morphometric measurements and blood values for the observed calves, are reported as useful information to implement protocols for bottlenose dolphin neonates’ husbandry management.

## 2. Materials and Methods 

This paper analyzes a structured veterinary program started in 2010 that allowed the collection of data related to bottlenose dolphin females during parturition, and postpartum data from 13 dolphin neonates during their first 30 days of life. The objective parameters analyzed were: timeline for specific events during the intrapartum phases, apnea duration, nursing frequency, morphometric measurements and blood parameters. 

These are reported as useful information to implement protocols for bottlenose dolphin calves’ husbandry management. 

### 2.1. Study Location

The three facilities included in the present study contain within their displays a nursery pool with a lifting platform where pregnant females are kept immediately prior to and during labor. The elevation of the bottom is mechanical and allows prompt intervention; in all 3 facilities, the false floor takes up to a maximum of 10 min to reach water surface. Nursery pool shape, dimension, depth and volume of water are reported in Table 1.

All the displays are provided with observation rooms with underwater viewing windows to facilitate direct observation of the dolphins. Visual observations were carried out directly by researchers. Recording surveillance webcam systems were installed for real time recording and used only in case of need. 

### 2.2. Mothers 

A total of 13 mother–calf dyads were considered for the present study (Table 2). Calves were numbered sequentially from C1 to C13, whilst females were named M followed by the number of their first calf.

Between June 2009 and December 2018 across the three facilities, eight females were diagnosed pregnant with serum progesterone determination [22] and by ultrasound examination during their first trimester [23]. The status of first time mothers (primiparous) for the wild born females M2, M5 and M6 was previously established by Biancani et al. (2009) [24]. All dolphins maintained in these facilities are trained to voluntarily participate in veterinary and husbandry procedures in order to guarantee regular routine diagnostic analysis [25]. The mothers were trained for voluntary venipuncture presenting the fluke for blood collection, without physical restraint. Complete blood count (CBC), biochemistry profile and hormones [26,27] were part of the diagnostic screening. To confirm the pregnancy status of each female and to estimate the stage of gestation, weekly ultrasonography was performed, utilizing portable ultrasound machines (Logiq-GE©) and 3.5 MHz convex probes. These regular exams were performed under voluntary training to follow up on the course of gestation and supervise the vitality of the fetus. Date of birth estimation was carried out as described by Lacave et al. (2004) [23]. During the last month of gestation, ultrasound examinations were performed daily to monitor the fetal heart rate [28]. 

Daily rectal temperature measurements [29,30] were recorded by inserting a protected 4 mm digital thermometer (Digitron 2000s, Digitron, Margam, Port Talbot, UK) probe approximately 10–12 cm in the rectum for an average of 2 min. Intermammary gland distance was measured with a caliber, considering the cranial and middle point distance between left and right mammary gland opening [31,32]. All the data collected were added in the set of prenatal information to predict the parturition. These data were used to decide whether the females needed to be separated or be kept within the group until just prior to labor. At OLT, the females were kept together with other females until rupture of the amniotic sac was observed. In the other two facilities, pregnant females were separated from the group for a maximum of two days before the expected delivery day.

Dedicated observers recorded all information regarding birth events and timing by monitoring the pregnant females for six hours every day (from 00.00–06.00 h) for up to 15 d before the estimated date of delivery. These observations aimed to identify the expected increase in abdominal flexions on the week before birth as reported by Tavolga and Essapian (1957) [33], Kinoshita et al. (1999) [31], and Joseph et al. (1999) [34]. 

All the above individual data collected, together with specific profile of the females, their maternal style, details on prepartum and stage 1 observations are not individually described in the present study, as this paper focusses on the outcome of postpartum events.

In essence, Robeck et al. [35] described parturition stages in bottlenose dolphins by identifying 3 different stages of periparturient parameters in the females entering labor: Stage 1: Labor signs with vaginal discharge, milk discharge, decreased appetite, contractions, decreased basal temperature, increased intermammary distance;Stage 2: Starts with the appearance of the flukes (podalic parturition represents 98% of the cases) and finishes with the birth of the calf;Stage 3: Complete expulsion of the placenta.

For the purpose of this paper, only objective parameters and timeline for specific events during the intrapartum phases (stage 2 and 3) for the individual females are reported as useful information to implement protocols for bottlenose dolphin calves’ husbandry management. 

### 2.3. Neonates

Observations of apnea intervals, nursing frequency, neonatal blood parameters and morphometric measurements were collected, analyzed and reported. Direct observations of mother and calf dyad were performed 24 h/day, from birth onwards, by the staff at the facilities, for up to the first 30 days of the calves’ lives.

The protocol established that in all facilities, the duration of apnea (time in between respirations) was recorded for 15 consecutive minutes every hour per observation session (counted in seconds with a stopwatch DT 2000, Digi instruments, Italy) from the pool side. 

The most relevant and immediate parameters to address in case emergency are hereby analyzed, such as: number of breaths within 5 min, average of apnea in seconds/hour (s/h), percentage of synchronic surfacing performed in dyad with the mother. 

Nursing activity was monitored at each facility through acrylic underwater windows. Every nursing event was recorded, including duration and evaluation of whether it could be considered successful or just an attempt. A nursing event that lasted a minimum of three seconds with a cloud of milk observed at the end of nursing act was classified as successful and was added to the data. Every unusual behavior and every potential disturbance (e.g., divers cleaning the pool) were also documented.

The data analyzed in this study extended to day 15 of the neonates’ life for apnea duration and nursing, to further compare Sweeney et al. (2010) [3], who stated that by day 7 to 10 of life data can be already evaluated to determine whether the development of a neonate is as expected.

Concerning physiological data collection (morphometric and bloodwork), the veterinary protocol established to preventively restrain and examine each calf on a regular schedule, at least for the first 30 days of life. In case of calves showing natural swimming, nursing and breathing without signs of abnormalities, the first capture and restraint for medical check-up was set on the second day of their life. Until 2015, if a calf did not nurse in the first 20 h after birth, intervention was considered within day one. After 2015, due to the experience with C1, this time frame was reduced to 15 h, without excluding the possibility of earlier intervention in case of manifested need. All calves were restrained on lifting platforms, allowing the dolphin mother to stay in close proximity to her offspring, providing reciprocal visual contact. All medical procedures (venipuncture for blood collection, nutritional support when necessary, antibiotic treatment and morphometric measurements) were completed with the calf immersed in the water. Neonates were only lifted out of the water for weighing on a regular basis through all the observation period.

Morphometric measurements were taken with a nonrigid measuring tape, to avoid any skin damage. Body length was measured from the tip of the rostrum to the notch of the tail fluke and the girth was measured 2 cm in front of the dorsal fin [36]. A SR500 scale (SR©Instruments, Tonawanda, NY, USA) was used to weigh the calves. 

With the calf still in the water and in close proximity to the mother, blood was collected from the periarterial venous rete in the ventral side of the flukes with 23 gauge butterfly needles for CBC and chemistry panel. Blood was then centrifuged (2500 rpm for 15 min) to separate the serum. Samples were taken to different analytical laboratories, depending on the facility. At pool side, when possible, a blood sample was examined for glucose as an indication of whether nutritional support was necessary. A portable glucometer (Accu-Chek, Roche Diagnostics, Mannheim, Germany) was used and a level of glucose <70 mg/dL was considered threshold.

When maternal nutrition was not enough, specific milk formula recommended for *Tursiops* neonates was freshly prepared every day and administered alone or as supplement to maternal milk, collected directly from the mother. Quantity and frequency of administration were established according to the growth rate and previous experiences [3,37].

### 2.4. Statistical Analysis

Statistical analysis was conducted to evaluate the factors that could have any type of influence and/or interference over the nursing pattern and apnea duration (intended as interval between breaths), both during the first 12 h and the first 15 days postpartum. For that purpose, Statgraphics^®^ Centurion XVI (The Plains, VA, USA) was used. Boxplots were used to represent the suckling activity during the first 15 days, showing median, lower and upper quartile. The whiskers indicate the minimum and maximum values of s/h and have a length of 1.5× IQR.

Considering only the first 12 h after birth, apnea measured in second per hours (s/h) and the natural logarithm of suckling (s/h) complied the requisites of a normal distribution (Kolmogorov–Smirnov Test: 21.55 ± 5.54, *p* = 0.546; and 1.38 ± 0.37, *p* = 0.547, respectively). In these cases, ANOVA and Fisher’s LSD (Least Significant Difference) test were used.

Dependent variables for the first 15 days, apnea (s/h) and suckling (s/h), did not comply the requirements of a normal distribution (Kolmogorov–Smirnov Test: 22.68 ± 5.33, *p* < 0.001; and 12.56 ± 15.68, *p* < 0.001, respectively) thus nonparametric tests were utilized, specifically Mood’s Median Test to compare the medians of two samples, and Kruskal–Wallis Test for more samples.

To evaluate statistical association between variables, Pearson’s and Spearman’s Correlation Coefficients were used for normal and non-normal variables, respectively. 

## 3. Results

### 3.1. Parturition

Following Robeck et al. [35] criteria and focusing on the events from birth onwards, all 13 calves monitored were born tail first as normally described physiologically for the species [22]. Five of these females were primiparous and their neonates overcame the observation period of 30 days, analyzed in the present paper. Three calves, born from multiparous mothers, did not survive the neonatal period (C1, C2 and C3).

Stage 1 was completely observed and data recorded only from M1 and the two birthing events of M4. The time between vaginal discharge of cervix mucous and water breaking (breaking of the amniotic sac) was registered (315, 510 and 1125 min, respectively) and the time between water breaking and delivery of calf was 75 min for C1, 540 min for C4 and 585 min for C8. With M3, M5 and M6 vaginal discharge of cervix mucous was not observed, but water breaking in the three females occurred 165, 120 and 190 min (for C7) before delivery, respectively.

Calves were born at different times of the day without showing any particular circadian or specific hour preference. The average duration of stage 2 of parturition was approximately 80 min (range 20 to 225 min) and 325 min for stage 3 (ranging from 220 to 570 min).

The time frame (in minutes) for stage 2 and 3 of labor, together with registration of first effective nursing (which is considered of utmost importance from a clinical perspective) are reported in Table 3.

Calf 1 and 2 died 19 and 58 h postpartum, respectively. In both cases, the animals were observed to breathe and nurse abnormally, as is reported in the nursing and apnea results. Calf 3 successfully nursed for the first time (at) 510 min after birth. However, it died on day 18 due to a traumatic injury.

### 3.2. Apnea Duration

Analysis of recorded data on apnea duration for the first 12 h of life showed a significant difference between the length of apnea during the first hour postpartum (16.04 ± 3.81 s/h) and the apneas during the period 3rd–12th hour (22.32 ± 5.47 s/h) (ANOVA, F = 1.94; *p* = 0.0395) as shown in Figure 1. Notwithstanding, considering only the neonates that died, no significant differences were observed in the same period of time (ANOVA, F = 0.51; *p* = 0.8764). 

The comparison of the apnea of the calves that survived to that of those that died (alive: 22.67 ± 5.47 s/h; dead: 17.25 ± 3.43 s/h), showed that apneas recorded during the first 12 h of life were significantly and statistically shorter (ANOVA, F = 29.02; *p* < 0.001) in the latter group (C1, C2, C3). 

There is a pattern of increase in the apnea duration that is lower in the calves that died. In fact, these animals presented an increase of just 3.61 ± 5.51 s between the first hour and the period 3rd–12th hour, and 5 ± 7.41 s between the first and the third hour of life. The data regarding the calves that survived showed an increase of 6.81 ± 4.84 s, and 5.81 ± 3.56 s for the same periods, respectively.

There were significant differences between the apnea during the first 12 h after the birth and the rest of the hours until day 15, being the apnea during the first 12 h of life significantly shorter (Chi square = 4.38; *p* = 0.0364). Data analysis showed significant differences in apnea duration (Kruskal–Wallis Test, H statistic = 1712.57; *p* < 0.001) among animals (N = 13) during the period 0–14 days of life.

The duration of the apnea in seconds/hour (s/h) among the 3 facilities was significantly different: ADG 24.11 ± 6.43 s/h, MMP 20.92 ± 3.89 s/h and OLT 24.99 ± 5.65 s/h (Kruskal–Wallis Test, H statistic = 395.72; *p* < 0.001). 

Considering all the animals (N = 13), the respiratory rate (RR) within the first 15 days of life ranged between 7 and 29 breaths/5 min (14.04 ± 3.29). Comparing the same time frame, the RR of animals that died during the observation period (N = 3) ranged between 9 and 27 breaths/5 min (15.35 ± 2.7) while in calves that survived (N = 10) the RR was 7–29 breaths/5 min (13.91 ± 3.31).

Synchronicity of apnea within the dyad was considered for eight of the calves during the first 24 h of life. Comparing clinically healthy animals (C8, C9, C10, C12 C13) to those that died or requested medical intervention (C1, C2, C11), the percentage of apneas performed by calves with mother was higher in healthy animals than in the second group (healthy: 92.12 ± 6.76%; problematic: 57.98 ± 16.95%).

The same analysis of the synchronicity of apnea within the dyad for five calves (C8, C9, C10, C11, and C12) during the first two weeks of life was performed with the data available. On day 15, the healthy (N = 4) calves were breathing in synchronicity with the mother for 74.50 ± 14.74% of the time. C11 showed a similar trend during the observation period but with lower percentage of synchronicity (on day 15th of 41.98%).

### 3.3. Nursing Activity

Nine calves nursed regularly during the observation period and first effective nursing act was recorded between three and 18 h after birth with an average of 10.86 ± 0.18 h postpartum, as reported in Table 3. On average, calves of primiparous females (N = 4) took longer to start nursing (13.5 ± 5.12 h) than those from multiparous mothers (N = 6, 9.1 ± 4.85 h). Calves born from Mother 4 (M4) were observed to start nursing before placenta was expelled. M8 started to show aggressive behaviors with the calf immediately after delivery and the calf (C11) was not able to nurse properly. Thus, C11 was restrained for the first time at 14 h of life for medical check-up and tube feeding.

By recording the duration in seconds of effective suckling, it was possible to determine the total suckling time per hour (s/h) during the entire observation period. All the neonates showed an individual pattern of suckling and the suckling activity differed significantly between animals (F = 14.35, *p* < 0.001) among the first 12 h. Out of the 13 calves observed, only five calves nursed during the first 12 h of life. Four of them (C3, C4, C8, C13) showed similar nursing behavior, whilst C5 statistically differed from the others.

When considering the nursing activity during the first 12 h of life, the five animals that started to eat within these hours showed a significant difference in suckling session length (ANOVA, F = 6.50; *p* = 0.0176) in regard to calves born at OLT and MMP, which spent more time suckling. Calves born at ADG are not considered, none started nursing before the 12 h of life. Among the remaining animals, five started suckling later (C6, C7, C9, C10, and C12) and three (C1, C2, and C11) were involved in the following difficult situations:-C1 showed two effective suckling events 17 h after birth, although it died two hours later.-C2 did not nurse and died on day 3.-Calf 11 was born from primiparous mother. Despite several attempts from the calf to nurse, the unexperienced mother did not accept any approach and even displayed aggressive behavior towards the calf. Due to lack of effective nursing events, the neonate was restrained for the first time at 14 h of life for a medical check-up and forced feeding. Colostrum was collected from the mother (38 mL) and administered by gastroesophageal tubing (9 mm diameter and 75 cm length, 50 cm mark) to Calf 11. During the following days, several attempts were conducted to allow the mother to take control of feeding the calf. However, the female was uncomfortable and still acting aggressively towards the calf, not allowing nursing to occur. Daily captures (from eight tube feeding sessions/day to two tube feeding sessions/day) were performed on Calf 11 for the first 24 days of life in order to ensure proper caloric intake, administering both maternal milk and formula.

The analysis of the data regarding the suckling activity per hour shows a steady increase in the suckling time during the first 24 h of life. From day one to day four, the calves spent more seconds/hour suckling (17 ± 20.3 s/h) than from day five to day 14 (11.3 ± 12.4 s/h), as shown in Figure 2. 

There were evident differences among animals when comparing the range of suckling per hour during the whole observation period. On day one, it ranged from 3 s/h to 160 s/h. On day 15, animals nursed from 4 s/h to a maximum of 95 s/h. In minutes per day, the nursing events considered for healthy animals (C4, C5, C6, C7, C8, C9, C10, C12, C13) reached a maximum of 7.2 ± 3.4 min/day on day 2 (N = 9), 3.2 ± 2.3 min/day on day 10 (N = 9) and 4.0 ± 3.6 min/day on day 15 (N = 7). However, calves of primiparous mothers presented statistically higher length (s/h) of nursing activity (Mood’s Median = 12.54; *p* < 0.001). 

No significant correlation was observed between the calves’ apnea and suckling duration, both during the first 12 h of life (Pearson’s r = 0.089; *p* = 0.089) and during the first 15 days (Spearman’s ρ = −0.029; *p* = 0.078).

### 3.4. Growth and Body Weight

Morphometric measurements (length, *n* = 67; girth *n* = 69) and body weight (BW) values (*n* = 86) were recorded when possible during medical interventions of the neonates and used to evaluate positive growth during the first 30 days of life. All values (*n* = 222) are reported on Table 4 grouped in intervals of time corresponding with the first maneuvers to handle the animals (Day 1–4), the following days up to day 10 of life (Day 5–10), and then clustered in intervals of ten days to cover the observation period (intervals Day 11–20 and Day 21–30, respectively). Only data collected on day 1 of life were considered for C11 in order to avoid influencing the results by including a hand fed animal with natural feeding patterns.

Considering the first and last BW, length and girth recorded for those animals with sufficient data (N = 9), the daily growth and increase in body weight was calculated for each individual. These values were used to determine the average daily growth and BW value for the group. In average, the neonates grew 0.521 ± 0.334 cm/d in length, 0.563 ± 0.177 cm/d in girth, and 0.428 ± 0.102 Kg/d. Figure 3 summarizes the recorded BW for each of the nine animals with sufficient data to calculate their average daily growth during the first 30 days of life. Data on C11 were not included as the animal was supplemented with artificial formula, described by Townsend [34].

### 3.5. Blood Parameters

As described in the introduction, this work is the result of the application of a practical veterinary protocol designed to overcome unwanted neonatal deaths. It therefore aims at supporting physical intervention wherever it could serve as a possibility to improve survivability of the calves. In this sense, 98 blood samples were collected. 

The protocol included the administration of prophylactic long-lasting antibiotics during the first capture for medical check-up immediately after the first blood sample collection. Thus the pretreatment results were examined in a separated group (0–Tx, *n* = 12, sample = 12, excluding C1). Parameters were monitored prioritizing clinical evaluation and sample volume availability in the following sequence: complete CBC profile, glucose, iron, BUN, total protein, ALT, AST, GGT, creatinine and electrolytes.

The 19 samples from animals that were clinically ill on the day of sampling or received extra medication prior to the sampling have been excluded from statistical analysis in order to be able to present a complete panel from healthy calves only. The blood samples have been divided in 4 groups shown in Table 5 according to the age of the calves, also covering from the blood draw after the prophylactic treatment until day 7 of life (Tx–7, *n* = 6, sample = 18), from day 8 to day 14 (8–14, *n* = 7, sample = 19), and from day 15 until day 30 (15–30, *n* = 7, sample = 30).

Results and values in Table 5 are presented as the 25% and the 75% quartiles around median for each considered period. 

## 4. Discussion

In the last two decades, a long-term commitment by veterinarians and animal care staff was devoted to extensively monitoring dolphin neonates’ health. Early intervention with advanced diagnostic and therapeutic techniques has proven to be essential to increase calf survivability [3,38,39].

This paper compares for the first time the outcome of 13 dolphin births in 3 different European zoological parks that applied a shared and standardized protocol. Although this absolute number might be limited in comparison, for example, with domestic animals, it represents 15.29% of the births that occurred in all 28 European dolphin facilities in those years (pers. communication R. Gojceta—EAZA *Tursiops truncatus* EEP Coordinator). 

For the purpose of this paper, all females were considered to be in stage 2 after observation of vaginal discharge (as described by Robeck et al. 2001) [22] and when flexions frequency occurred at a minimum of 10 per hour (defined by Krames and Krames (1996) [40] as a prodromal signal of parturition). All the mothers, including the ones whose calves died, delivered within 4 h (maximum 225 min), and expelled the placenta within 12 h postpartum, confirming Robeck et al. 2018 [35]. M4, who delivered healthy calves, had the longest periparturient time. 

The duration of each single event proves to be useful and needs to be carefully analyzed and overlooked with experienced clinical observations.

The monitoring of respiratory activity is also considered essential to establish the state of neonate health [3]. In this study, the animals that died earlier showed significantly shorter duration of apnea than those surviving. Interestingly, the retrospective statistical analysis showed that C3, who was a female that died 18 days postpartum, appeared to be a healthy animal and had length of apnea similar to those that died within the first three days. According to morphometric measurements of the fetus registered during the pregnancy [23], C3 was born three weeks before the expected day of delivery. It was hypothesized that it could have been premature and, thus, had a more delicate immune system or a weaker physical constitution. However, female calves have an as yet unexplained higher neonatal mortality rate than males [12].

In general, the first hours of life are crucial for the survivorship of a dolphin calf, and for a clinician it is fundamental to have as much information as possible to decide whether or not to intervene. As previously observed by Peddemors in 1990 [41], the calves in the present study show an increasing pattern in apnea duration in the first 12 h. In addition, a second step of positive breathing capacity development occurs when apneas become longer during the first 15 days of life. However, the results show higher variability in the apneustic plateau (prolonged pause between inspiration and expiration) in the animals that died than in the calves that lived. The minimum increase in the duration of apnea in animals that survived was 2.25 s between the first and third hour of life. Values below that level of increase were only observed in animals that died. Therefore, this suggests that those calves that do not achieve an increase in apnea equal to or greater than 2.25 s during the first 3 h of life may require medical intervention.

The respiratory rates of animals herein monitored were within the limits by Sweeney et al. [3], who reported that RR is also an important tool to evaluate the neonatal health status, considering 12 to 25 respirations/5 min (up to 30) as normal. The observations performed from the first hours to the first 15 days of life were carefully used to identify early signs of clinical problems, and to get indications for medical intervention. Although the calves’ average apnea duration was within the normal limits reported by the literature, the present study identifies a significant difference when this parameter is correlated to the location (size and shape of the pools), as previously observed by Kleiva [42]. Wider pool sides may prolong the time needed to cover a straight underwater horizontal path, thus keeping longer apneas before surfacing to breathe. 

The percentage of surfacing performed by the dyad in synchronicity confirmed the data reported for healthy animals in the literature [7,41,43]. The capability of a calf to swim in synchronicity with its mother is more dependent on the individual’s behavior rather than other external factors, such as the environment [42].

The results of this study show that successful calves started to nurse between 3 and 18 h after birth. Several authors agree that the observation of nursing activity is fundamental in order to monitor the wellness of bottlenose dolphin calves, together with social and behavioral development [3,38,44,45]. Based on the literature, nursing activity generally begins within the first 12 h postpartum [3,35,44,46], which suggests that an intervention is necessary if the calf has not been observed to nurse within the first 24 h [3,38]. On average, the present data showed that there is a tendency for the calves of primiparous female bottlenose dolphins to take longer to start nursing than those from multiparous mothers, as previously suggested by Sweeny et al. (2010) [3] and von Streit et al. (2013) [47]. Thus, medical restraint may become more advisable if a calf does not successfully latch on within the first 20 h of life. There is a natural trend related to the increase in suckling efficiency as the calf grows older [45,47], and a peak in mother´s milk production as well as changes in milk composition [45,46,48]. In alignment with previously reported observations [3,38,44], an increase in suckling per hour was also observed in this study during the first four days postpartum, followed by a remarkable decline starting from day 5. In addition, the calves of primiparous mothers presented higher length (s/h) of nursing activity, as previously reported [47]. 

The collection of morphometric measurements and body weight, together with visual inspection of body shape, are essential parameters to indicate what is “normal” in the development of a calf [3,38]. Sweeney et al. (2010) [3] reported that a normal neonate should gain approximately an average of 0.2 kg a day. Our results registered an increase of approximately 0.4 ± 0.1 kg/d, possibly related to more frequent morphometric data collection or other factors (i.e., seasonality, location or milk quality) that would need to be further investigated. Daily growth was determined by measuring the increase in length and girth, and were, respectively 0.521 ± 0.334 cm/d and 0.563 ± 0.177 cm/d. These represent the first set of progressive growth data available for neonates within the first 30 days of life. 

Concerning bloodwork, the general interpretation of the results is a very important diagnostic activity [49] that requires a deep evaluation of each parameter within the context, thus allowing thorough analysis from a clinical perspective by an experienced veterinary clinician. 

A set of hematology and blood chemistry values are presented in this paper aiming to stablish baselines references for clinically healthy bottlenose dolphin neonates (<30 days). These type of references has been long accepted and used [27]. Thus, the results on Table 5 represent a great addition to the experiential knowledge. Interval blood values in healthy neonate bottlenose dolphins have been previously reported only by Sweeney et al. (2010) [3]. Flower et al. (2018) [39] report the only blood reference available in the literature for day 1 dolphin neonates, although it was from a calf requiring critical care and hand rearing. Results by Flower [39] showed values altered due to hemoconcentration and lack of nursing, and would have been excluded from the present study that only shows healthy animals data. Thus, the results described in this study for the period 0–Tx can be considered as the first baseline reference available for clinically healthy animals during the first days of life. Sweeney et al. (2010) [3] reported average values for neonates on day 7, 15 and 30. The protocol here presented included more frequent medical check-ups, applied also to counteract previous losses. Therefore, more complete data between day 5 and 7, 8 and 14, and 15 to 30 appear, broadening the clinical perspective and including results on parameters such as cholesterol, triglycerides, α-amylase, lipase, magnesium and cortisol.

Concerning CBC, the erythrocytes values (RBC, HGB, HTC, MCV, MCH and MCHC) increase during the first 15 days and decrease mildly after the second week of life. The general ranges are comparable to those described for wild Atlantic adult bottlenose dolphins [50]. This could be coinciding with the improved breathing competence that occurs around 15 days of life [41]. Furthermore, the percentage of reticulocytes increases steadily and reaches higher levels than dose previously reported [51] during the period 15–30 days. 

The amount of circulating leucocytes (WBC) increases slightly during the first week of life and stabilizes. WBC values for animals above seven days of life are within the ranges reported for captive dolphins (SeaWorld) by Gulland et al. (2018) [51]. There is a rapid increase in the percentage of monocytes after the prophylactic treatment. These levels decrease after the first week of life. Furthermore, a steady increase in eosinophils and platelets is evident during the 30 days. The range of platelets available in the literature for captive [51,52] and wild Atlantic bottlenose dolphins [27,48] is lower than the values presented in this study after the first week of life. 

The blood chemistry panel considered normal for these healthy calves can serve as reference for future studies. 

Glucose values increase over the studied period, still remaining within the reported parameters in *T. truncatus* under human care [51], and are comparable to those from their wild juvenile counterpart [27]. However, neonates have higher glucose levels than wild adult individuals [50].

Only the reference values reported for dolphins at SeaWorld [51] regarding BUN levels are comparable with the results for neonates older than a week. BUN levels increase during the first week and steadily decrease in the following weeks. Creatinine levels remain stable during the first 30 days of life, although they appear lower than the reported levels [27,50,51]. In the period studied, the neonates present a constant increase in ALP which is higher compared to the literature available [27,50,51]. However, the ALT, AST, GGT and iron values only increase during the first week of life. Then, the levels decrease steadily. ALT and AST within the first seven days are comparable with the references by SeaWorld [51] and Goldstein et al. [50], respectively. The values for CK decrease from the second week of life onwards and are higher than in the references [3,27,50,51].

The neonatal total amount of proteins remains fairly stable during the first 30 days, although with lower values compared to the ones reported for free ranging bottlenose dolphins [50,51] and the ones under human care [51,52].

Cortisol has been used and recognized as a biomarker of stress response [53,54,55,56,57,58], and has relevant influence over the health of animals [2]. Whilst capturing and handling dolphins are considered as stressors, no values of cortisol in dolphin neonates have been previously reported. Considering the results for this hormone from the neonates’ first capture (0–Tx, 1.72 ± 0.64 µg/dL), similarities are found with those reported by St. Aubin et al. (1996) [54] for the semidomesticated adult dolphins. The values available for free-ranging dolphins at capture are higher [54,55,59]. The analysis of the results for the other three periods considered in the present study showed retrospectively a decrease in the serum cortisol. These values could be hypothesized as habituation response to capture and handling procedure. Besides, cortisol levels for the period 15–30 days of life are similar to those reported for peripubertal dolphins [55]. 

The values obtained on sodium, potassium and the Na:K ratio are also comparable to those reported by Ortiz and Worthy (2000) [59] and Hansen and Well (1996) [60], suggesting a state of ionic and osmotic homeostasis and a lack of capture stress. Furthermore, this could corroborate the fact that neonates get used to the handling, as suggested by Sweeney et al. (2010) [3]. 

The above confirms that blood samples are among the most relevant diagnostic tools with collection dolphins [49].

## 5. Conclusions

The intrapartum and postpartum data here reported provide a large set of information that could be useful for the early detection of health issues in newborn *Tursiops truncatus*. 

Careful analysis of the length of the second stage of parturition with experienced clinical observation is advised. Medical handling of the calf for physical examination, body weight recording and blood work is recommended at 48 h. The cortisol blood level should decrease after the first week and BW should increase at least 0.428 ± 0.102 kg per day. 

In addition, careful monitoring of the respiratory activity of the calves is of utmost importance. Average apnea duration, in correlation with the size and shape of the pools, and modifications of the capability of the calf to swim and breathe in synchronicity with its mother, are essential to detect any underlying health issues. 

Larger horizontal dimensions of the pools influenced the apnea timeline. This could be related to the swimming pattern of the dyad that could allow a different control of the calf by the mother and allow the calf to swim attached to her in a straight line for a longer period of time. Future studies could also investigate the synchronicity of the swimming pattern, as well as correlations between the maximum length of apneas and the pools’ dimensions. 

The most remarkable situations that might induce an early proactive intervention are:If the calf does not present an increase in apnea duration of at least 2.25 s between the first and the third hour of life.If the placenta is not expelled within 12 h after the birth of the calf.If the neonate does not successfully latch on within 20 h postpartum.If the percentage of synchronic surfacing by the dyad does not reach an average of 92.12 ± 6.76%.If any clinical signs of disease are observed.

The compilation of the data here described represents supplementary indicators of husbandry and management conditions included within the five domains model by Mellor in 2017 [61], and constitute a valid contribution to guarantee a positive welfare state of bottlenose dolphin neonates under human care.

## Figures and Tables

**Figure 1 animals-11-01066-f001:**
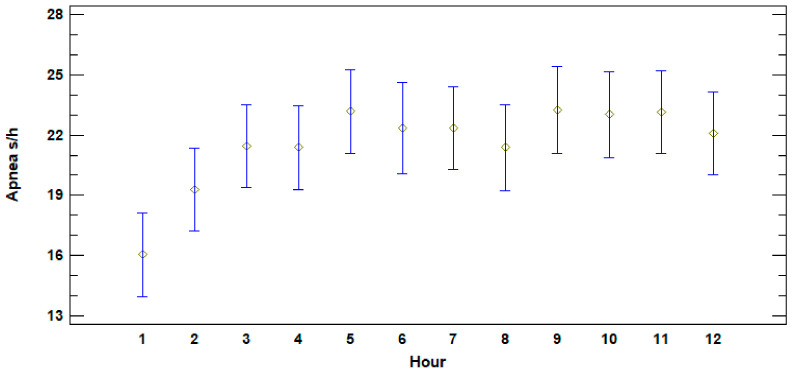
Average apnea duration in seconds per hour of calves during the first 12 h of life (*n* = 13). The Y axis shows apnea duration in seconds per hour and the X axis shows the hours from birth.

**Figure 2 animals-11-01066-f002:**
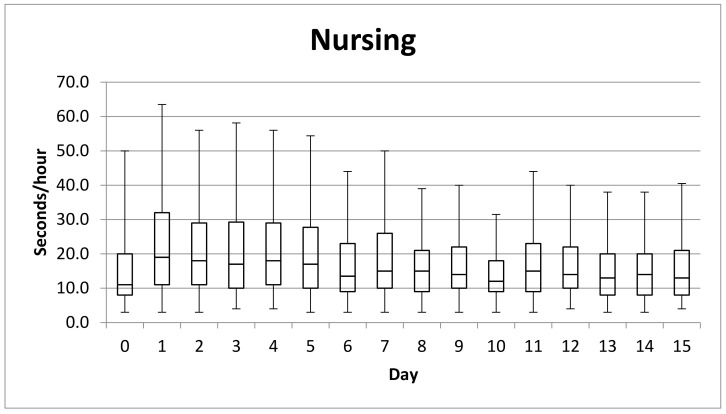
Median of suckling during first two weeks. Nursing events of less than 3 s were not considered. Boxplot shows the quartiles and outliers of average suckling time per hour. The whiskers have a length of 1.5xIQR and represent the maximum and minimum nursing time spent by the calves per hour.

**Figure 3 animals-11-01066-f003:**
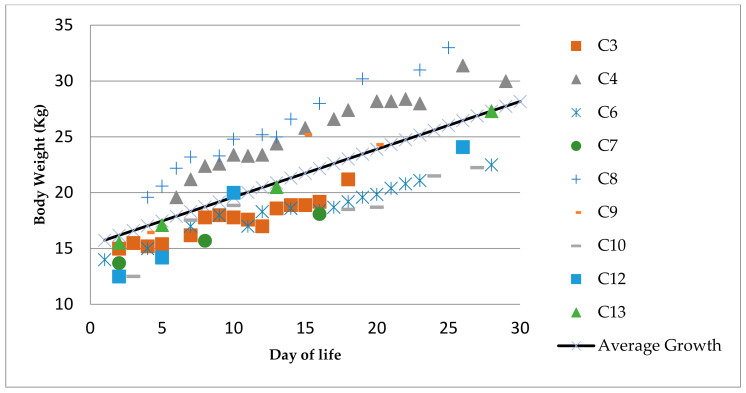
Trend of weight gain per day during the first 30 days of life (*n* = 9). Body weight is represented on the Y axis while days of life are indicated on the X axis.

**Table 1 animals-11-01066-t001:** Characteristics of the nursery pool across facilities including shape of the pool, water surface area, maximum depth and water volume.

Facility	Shape	Water Surface Area (m^2^)	Maximum Depth (m)	Water Volume (m^3^)
Mediterraneo Marine Park	Circular	78	3.5	432
Oltremare	Rectangular with rounded edges	240	3	850
Acquario di Genova	Rectangular with rounded edges	198	4	1000

**Table 2 animals-11-01066-t002:** Details of the mothers and their respective calves included in the study. None of the mothers are related. The table shows the ID of the mothers, their country and year of birth, whether they had previous parturition, their respective calves and the European facility where birth occurred.

Mother	Country and Year of Birth	Previous Parturition	Calf	Facility
M1	WB (USA) 1979	YES	C1	OLT
M2	WB (CUBA) 1997	YES	C2	MMP
M2	WB (CUBA) 1997	NO	C5	MMP
M2	WB (CUBA) 1997	YES	C13	MMP
M3	CB (ITALY) 1997	YES	C3	OLT
M4	CB (ITALY) 1994	YES	C4	OLT
M4	CB (ITALY) 1994	YES	C8	OLT
M5	WB (CUBA) 1998	NO	C6	MMP
M6	WB (CUBA) 1999	NO	C7	MMP
M6	WB (CUBA) 1999	YES	C10	MMP
M6	WB (CUBA) 1997	YES	C12	MMP
M7	CB (ITALY) 2001	NO	C9	ADG
M8	CB (ITALY) 1995	NO	C11	ADG

WB: Wild born; CB: Captive born; OLT: Oltremare, Italy; MMP: Mediterraneo Marine Park, Malta; ADG: Acquario di Genova, Italy.

**Table 3 animals-11-01066-t003:** Time frame in minutes of periparturient parameters for each birth considered in the study. The length of Stage 2 comprises the appearance of the flukes to the birth of the calf. Stage 3 commences at birth of the calf and finalizes with the complete expulsion of the placenta.

Mother ID	Calf ID	Calf Sex	Date of Birth (Day/Month/Year)	Time of the Birth	Length of Stage 2 (Minutes)	Length of Stage 3 (Minutes after Birth)	First Effective Nursing (Minutes after Birth)
M1	C1	Female	19/01/2015	16:43	43	285	n/a
M2	C2	Male	13/08/2016	01:37	37	220	n/a
M3	C3	Female	31/05/2010	04:20	105	300	510
M4	C4	Male	02/07/2010	08:30	150	265	240
M2	C5	Male	20/07/2010	03:21	60	290	390
M5	C6	Male	08/08/2010	19:05	100	360	990
M6	C7	Male	20/12/2010	14:51	60	360	1080
M4	C8	Male	09/08/2014	20:59	225	570	180
M7	C9	Female	01/09/2014	03:48	45	240	780
M6	C10	Female	31/10/2014	19:00	20	345	900
M8	C11	Male	20/08/2015	02:13	110	410	n/a
M6	C12	Male	09/12/2018	22:03	50	240	780
M2	C13	Male	01/10/2019	22:52	29	332	667

**Table 4 animals-11-01066-t004:** Body weight and morphometric measurements (length–girth) of the calves during their first month of life. Data are presented grouping the weight and morphometric measurements into 4 periods (Day 1–4, Day 5–10, Day 11–20, and Day 21–30) for better understanding and easier clinical reference.

	Day 1–4					
	N	Samples	Min	Max	Av	SD
Body weight (Kg)	11	14	12.5	20.6	15.23	2.36
Length (cm)	11	13	95.5	118	107.98	4.91
Girth (cm)	9	11	48.3	72.5	61.80	6.41
	**Day 5**–**10**					
	**N**	**Samples**	**Min**	**Max**	**Av**	**SD**
Body weight (Kg)	8	23	14.2	24.8	19.43	3.00
Length (cm)	6	19	106	117.5	112.59	3.55
Girth (cm)	7	20	61	74	68.30	4.04
	**Day 11**–**20**					
	**N**	**Samples**	**Min**	**Max**	**Av**	**SD**
Body weight (Kg)	8	33	17	30.2	21.83	3.90
Length (cm)	8	25	108	125	116.65	4.32
Girth (cm)	8	28	64	80.5	71.91	4.13
	**Day 21**–**30**					
	**N**	**Samples**	**Min**	**Max**	**Av**	**SD**
Body weight (Kg)	6	15	20.4	33	26.00	4.38
Length (cm)	5	10	112	131	125.50	5.33
Girth (cm)	5	10	69	85.5	78.55	4.57

**Table 5 animals-11-01066-t005:** Neonate hematology and blood chemistry of clinically healthy calves. Results and values are presented as the 25% and the 75% quartiles around the median for each considered period. The ranges cover from the blood draw after the prophylactic treatment until day 7 of life (Tx–7, *n* = 6, sample = 18), from day 8 to day 14 (8–14, *n* = 7, sample = 19) and from day 15 until day 30 (15–30, *n* = 7, sample = 30).

Parameter	0–Tx			Tx–7			8–14			15–30		
	*n* = 12, Sample = 12			*n* = 6, Sample = 18			*n* = 7, Sample = 19			*n* = 7, Sample = 30		
	*n*	25%	75%	*n*	25%	75%	*n*	25%	75%	*n*	25%	75%
**RBC** (10^6^/uL)	11	3.85	4.04	18	3.86	4.40	17	3.60	4.23	29	3.41	3.93
**HGB** (g/dL)	11	14.65	15.40	18	15.03	16.73	17	13.90	15.10	29	12.80	14.40
**HCT** (%)	11	43.80	46.5	18	43.13	46.30	17	41.00	43.20	29	37.40	41.70
**MCV** (fl)	11	112	115.05	18	105.53	109.83	17	102.10	113.00	29	107.80	111.30
**MCH** (pg)	11	36.60	39.30	18	35.10	39.68	17	35.70	39.10	29	36.00	38.60
**MCHC** (g/dL)	11	33.10	34.00	18	33.75	36.00	17	33.80	35.20	29	33.20	34.90
**WBC** (10^3^/uL)	11	4.12	7.40	18	3.44	7.10	17	5.09	8.40	29	5.20	8.00
**Neutrophils (band)** (%)	12	0.00	0.00	18	0.00	0.00	16	0.00	0.00	29	0.00	0.00
**Neutrophils (mature)** (%)	12	56.25	77.25	18	44.25	68.13	17	39.20	79.00	29	60.00	75.00
**Lymphocytes** (%)	12	18.50	28.25	18	16.73	24.93	17	13.00	43.50	29	16.60	23.00
**Monocytes** (%)	12	2.00	5.75	18	5.25	34.53	17	3.00	8.40	29	3.00	9.00
**Eosinophils** (%)	12	0.00	2.00	18	1.18	2.93	17	1.40	3.00	29	2.00	6.60
**Reticulocytes** (%)	0	N.D.	N.D.	2	1.27	1.36	1	3.00	3.00	8	5.46	6.49
**Platelets** (10^3^/uL)	11	119.50	169.00	17	157.00	218.00	17	187.00	242.00	29	327.00	419.00
**Glucose** (mg/dL)	9	91.50	130.45	16	92.00	121.25	16	116.50	150.88	22	128.00	148.75
**BUN** (mg/dL)	9	42.00	63.50	16	47.00	69.50	17	49.00	58.00	25	43.00	49.00
**Creatinine** (mg/dL)	9	0.50	0.70	14	0.50	0.71	16	0.55	0.70	25	0.68	0.80
**Cholesterol** (mg/dL)	5	174.00	210.00	10	217.50	248.00	11	219.81	278.10	16	208.25	262.26
**Triglycerides** (mg/dL)	5	221.30	328.00	10	151.00	238.75	11	116.50	237.50	19	137.00	200.50
**Bilirubin** (mg/dL)	8	0.32	0.68	12	0.28	0.43	11	0.05	0.20	19	0.10	0.28
**ALP** (U/L)	8	1825.50	2270.75	13	1699.00	2266.00	11	2343.50	4503.50	21	3250.00	5864.50
**GGT** (U/L)	6	22.25	52.75	10	45.00	58.50	11	43.50	49.50	20	32.00	37.25
**GOT-AST** (U/L)	7	172.50	207.50	12	164.00	225.25	16	144.50	193.75	24	127.75	179.50
**GPT-ALT** (U/L)	9	35.00	40.00	15	26.05	46.00	16	14.00	28.13	25	13.00	25.00
**α-Amylase** (U/L)	5	2.00	3.00	7	1.00	2.35	4	1.00	2.20	16	1.00	2.00
**Lipase** (U/L)	3	10.00	15.50	6	13.25	20.00	3	8.00	10.00	14	16.00	21.00
**CK** (U/L)	6	623.00	1182.75	10	595.50	1083.25	10	458.75	551.25	19	520.50	838.50
**LDH** (U/L)	6	753.00	1026.25	10	999.05	2245.25	10	697.50	898.23	20	533.75	1857.25
**Tot. protein** (g/L)	9	53.00	62.00	15	58.70	61.25	17	52.00	57.00	24	51.83	54.70
**Albumin** (g/L)	8	38.00	41.25	12	39.23	45.40	14	37.53	41.30	20	33.75	39.48
**Globulin** (g/L)	8	12.53	19.75	12	17.08	21	14	15.33	17.65	20	12.68	21.00
**Alb/Glob**	9	1.82	3.75	14	1.90	2.32	16	2.16	2.67	23	1.66	3.25
**Sodium** (mmol/L)	9	158.00	162.00	15	153.70	160.50	15	153.50	157.00	24	151.60	154.00
**Potassium** (mmol/L)	9	4.60	4.90	15	3.45	4.45	15	3.70	4.20	25	3.70	4.10
**Chloride** (mmol/L)	7	112.50	117.00	10	101.30	110.00	14	106.00	111.00	24	108.98	112.00
**Phosphorus** (mmol/L)	7	1.28	2.10	15	1.50	2.23	14	2.08	2.45	23	1.87	2.26
**Calcium** (mmol/L)	9	2.30	2.52	14	2.06	2.54	15	2.12	2.70	22	2.23	2.45
**Magnesium** (mmol/L)	4	1.14	1.40	8	0.65	0.82	12	0.60	0.81	17	0.70	1.00
**Fibrinogen** (mg/dL)	4	125.00	230.00	10	211.50	323.00	8	272.75	354.00	22	254.00	637.00
**Cortisol** (µg/dL)	5	1.40	1.90	6	0.63	2.00	9	0.40	1.00	18	0.30	1.00
**Iron** (µg/dL)	9	251.23	413.40	13	249.00	502.00	15	228.70	343.00	27	183.00	300.00
**ESR** (mm/h)	3	0.50	6.50	4	0.00	3.00	5	1.00	4.00	13	1.00	4.00

## Data Availability

Data included in individual animals’ clinical files are not published in public databases—data can be available upon request due to internal private company policies.

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
