# Peer review of "Physiological Parameters Monitored on Bottlenose Dolphin Neonates (Tursiops truncatus, Montagu 1821) over the First 30 Days of Life"

_animals, 2021, doi:10.3390/ani11041066_

Round 1

Reviewer 1 Report

Overall, the revised version of the manuscript provides a much clearer description of the results and is now easier to follow. Furthermore, the authors addressed the findings on the blood values and improved the discussion part considerably in their revised manuscript. However, there are few points regarding the results that I would like the authors to attend to:

a) The captions of the tables and figures have been improved substantially and are self-explanatory now. However, it is uncommon practice to put the name of the statistical tests performed into the captions of the figures as they do not represent the statistical analyses and are only the graphical presentation of the data sets. The statistical tests should be mentioned with the test output though. E.g., l. 324 “apnea’s during the period 3rd-12th hour (22.32±5.47 s/h) (ANOVA, F=1.94; p=0.0395) as shown in Figure A”. Please correct throughout the manuscript.

Minor corrections:

l. 406 Please double check, it seems the reference to Figure A is incorrect here

From line 444 on, the Figures’ labels seem to be incorrect, but that might be just due to some format issue as track changes are not presented correctly in some parts so that it is difficult to follow whether parts have been deleted or moved. Please double check.

Reviewer 2 Report

-The authors have addressed all comments raised by the reviewers. Thus, this reviewer would like to suggest that the manuscript is acceptable for the publication in the journal.

Author Response

This manuscript is a resubmission of an earlier submission. The following is a list of the peer review reports and author responses from that submission.

Round 1

Reviewer 1 Report

Comments in the attached document.

Author Response

sincerely

Claudia Gili

Reviewer 2 Report

In this research paper, authors discussed the Management of Neonate Bottlenose Dolphins 2 during the First 30 Days of Life. Author summarized the strategies to detect early health issue in newborn bottlenose dolphins. This manuscript includes the emerging approaches, which enable the readers to understand and follow the Management of Neonate Bottlenose Dolphins 2 during the First 30 Days of Life. Minor revision required. I have attached comment file. Please find the attached file.

Thank you.

Author Response

Please see attached letter and revised paper

sincerely

Claudia Gili

Reviewer 3 Report

The authors present long-term data collected from observations and medical examination of 13 dolphin neonates in three different European facilities over a period of 10 years.

Overall, I found the paper to make a valuable contribution to the field of marine mammal community and contribute to imrovement of animal welfare under human care. However, there are some aspects of the paper that I would like to see improved. In its current form, the manuscript lacks clarity and is difficult to follow.

My main concern is the presentation of the results. The result section is quite difficult to follow. In general, the Figures' and Table's legend are not informative and it is not clear what is shown. For example, Table 5 is presented without any further details and explanation of the results. The authors report the single blood parameters and deal with the findings in the discussion section but they do not really present the results of the blood chemistry.   

In general, I think clear goals would help a reader unfamiliar with the topic to understand what was done and why and providing more informative figure legends would make it easier to follow the results. Also, I suggest to state more specifically how the provided information can be applied to improve the husbandry and welfare of new born bottlenose dolphins under human care.

Author Response

Please see attached letter and revised paper

sincerely

claudia
